# On the Quest for In Vitro Platelet Production by Re-Tailoring the Concepts of Megakaryocyte Differentiation

**DOI:** 10.3390/medicina56120671

**Published:** 2020-12-03

**Authors:** Patricia Martínez-Botía, Andrea Acebes-Huerta, Jerard Seghatchian, Laura Gutiérrez

**Affiliations:** 1Platelet Research Lab, Instituto de Investigación Sanitaria del Principado de Asturias (ISPA), 33011 Oviedo, Spain; uo266133@uniovi.es (P.M.-B.); andreaacebeshuerta@gmail.com (A.A.-H.); 2Department of Medicine, University of Oviedo, 33003 Oviedo, Spain; 3International Consultancy in Strategic Safety/Quality Improvements of Blood-Derived Bioproducts and Suppliers Quality Audit/Inspection, London NW3 3AA, UK; jseghatchian@btopenworld.com

**Keywords:** platelet transfusion, megakaryocyte differentiation, in vitro platelet production, cell source, cell reprogramming

## Abstract

The demand of platelet transfusions is steadily growing worldwide, inter-donor variation, donor dependency, or storability/viability being the main contributing factors to the current global, donor-dependent platelet concentrate shortage concern. In vitro platelet production has been proposed as a plausible alternative to cover, at least partially, the increasing demand. However, in practice, such a logical production strategy does not lack complexity, and hence, efforts are focused internationally on developing large scale industrial methods and technologies to provide efficient, viable, and functional platelet production. This would allow obtaining not only sufficient numbers of platelets but also functional ones fit for all clinical purposes and civil scenarios. In this review, we cover the evolution around the in vitro culture and differentiation of megakaryocytes into platelets, the progress made thus far to bring the culture concept from basic research towards good manufacturing practices certified production, and subsequent clinical trial studies. However, little is known about how these in vitro products should be stored or whether any safety measure should be implemented (e.g., pathogen reduction technology), as well as their quality assessment (how to isolate platelets from the rest of the culture cells, debris, microvesicles, or what their molecular and functional profile is). Importantly, we highlight how the scientific community has overcome the old dogmas and how the new perspectives influence the future of platelet-based therapy for transfusion purposes.

## 1. Background: Platelet Transfusion over the Years

Platelet (PLT) transfusions were originally given in a therapeutic manner to those patients suffering from bleeding episodes, as it occurs in congenital thrombopathic patients or in patients suffering from acute hemorrhage caused by traumatic injury or hemorrhagic complications derived from surgery [1,2]. The use of PLT transfusions in a prophylactic manner, i.e., to prevent bleeding, has increased over the past years, and it is advised for those patients at risk of severe bleeding, as it occurs in onco-hematologic patients, whose condition and treatment induce a reduction and dysfunction of PLTs [3,4]. Over the last decade, there has been a steady, worldwide increase in the demand of PLT concentrates (PCs) for transfusion [5,6], which is concordant with the improvement of PLT disorder diagnosis and management and a growing, aging population with its concomitant increase in the incidence of onco-hematological diseases [7,8].

Currently, the source of PLTs comes uniquely from donations. The half-life of PLTs is short, thus their storage time or shelf life is limited to a few days [9]. They are generally stored shaking at room temperature in order to prevent loss of PLT functional integrity, which also poses a risk of bacterial or viral contamination [10,11]. While the implementation of pathogen reduction treatments (PRTs) has allowed an extension of the shelf life and a safer transfusion product, thus enabling blood centers to better manage their PC supplies, the demand is still causing pressure to find alternative solutions to increase the availability of PLTs for transfusion [12,13]. This is in part due to the well-known fact that long-term stored PCs start presenting signs of decay, the so named PLT storage lesion [14], which directly conditions the efficacy of PLT transfusions and seems to be enhanced by certain PRTs [15,16,17]. Briefly, the implementation of PRTs allows for longer storage times and a safer product, however, the tendency to provide hospitals with long-term stored PCs (in order to better manage the PC pool at blood banks) results in less efficient transfusions, which in turn calls for repetitive PLT transfusions per patient, all leading to an increase in the costs [18].

In this regard, efforts are being made at different levels in order to meet the demand of PLT transfusions. At the level of PCs, there are several studies focusing on the effect of the different PRT methods on the mentioned PLT storage lesion as to identify the ones that are compatible with a better preservation of PLT functionality. Other studies, for example, aim at the development of storage methods that would extend PLT lifespan in PCs without disturbing their functionality (either at 4 °C or cryopreserved) [19].

In parallel, the possibility to boost endogenous PLT production with exogenous or recombinant thrombopoietin (TPO), or TPO receptor agonists, has provided an alternative approach to PLT transfusion in certain patients or clinical situations. Nevertheless, this therapeutic approach is incompatible with the set of patients with higher PLT transfusion demands (i.e., onco-hematological), rendering transfusions non-substitutable in terms of efficacy and even safety [20]. 

In this review, we focus on the efforts put into producing PLTs in vitro, which involves the growth and the differentiation of primary megakaryocytes (MKs) or megakaryocytic lines, with the aim to develop current good manufacturing practices (cGMP)-grade PLTs in a large-scale manner, with the ultimate goal of overcoming the aforementioned shortage of PCs from donations. We recently reported the variables that we think are important to consider in this regard [21], and in the present review, we expand on those issues, since we believe they are not always acknowledged despite their tremendous impact on the desired functional PLT product.

## 2. In Vitro Megakaryocyte (and Platelet) Production: The Story Thus Far

### 2.1. Thrombopoietin and other Factors Influencing Megakaryopoiesis

PLTs are the smallest anucleate components of the blood and play a capital role not only in the maintenance of hemostasis but also in angiogenesis, cancer, embryonic development, inflammation, and immunity [22]. They derive from bone marrow (BM) MKs, who undergo a complex differentiation process driven mainly by the actions of the hormone TPO [23].

The discovery of TPO opened the door to not only the development of therapeutic approaches (as evidenced by the use of TPO receptor agonists of first and second generations in several pathologies of different etiology to induce PLT production in patients, thus avoiding the use of purified or recombinant human TPO), but also to the possibility to culture MKs in vitro [24,25]. The first proof of TPO as an MK stimulating factor was obtained from colony-forming experiments, which showed that TPO alone produces near maximal numbers of MK colony-forming units (as compared to other mixtures using other growth factors required for the expansion of early hematopoietic precursors, such as stem cell factor (SCF)) [26]. However, the TPO receptor, MPL, is not restricted to the MK committed precursor cells but is also expressed in hematopoietic stem cells (HSCs) [27,28], where it contributes to the regulation of their self-renewal, proliferation, and differentiation, in addition to its actions on MK committed progenitors [29,30,31]. That is why, when used in in vitro culture, and depending on the source material, it is frequent to obtain a highly heterogeneous cell culture enriched in MKs at different stages of differentiation as opposed to other hematopoietic lineages that can be grown in a highly pure and synchronous manner when adding the specific growth factors (erythroid cultures, B-cells, T-cells, neutrophils, macrophages, etc.) [32,33].

Several studies have pinpointed the role of TPO at the early, but not terminal, stages of MK maturation, where other factors such as stromal cell-derived factor 1 alpha (SDF1α), fibroblast growth factor 4 (FGF4), and sphingosine-1-phosphate (SP1) might play a role [34,35,36]. Furthermore, there is still residual megakaryopoiesis in Tpo-deficient mice, suggesting the existence of alternative, TPO-independent mechanisms of MK generation [37]. In addition, there seems to be a difference in the processes driving steady state MK differentiation as compared to the process of MK differentiation and PLT production under subjacent inflammation, which has been recently unraveled, and where even PLTs display a distinct functional profile [38,39].

Nowadays, a number of MK culture methods still consider the addition of several of these factors, along with a variety of cytokines. Some of the most common ones include members of the β common cytokine family or colony-stimulating factors (CSF), such as granulocyte-macrophage CSF (GM-CSF) and interleukine (IL)-3, and of the IL-6 family (IL-6 and IL-11), IL-9, erythropoietin (EPO), and cytokines produced by BM stromal cells, such as SCF and/or FMS-like tyrosine kinase 3 ligand (Flt3-L) [40,41].

Gene knockout mouse models have shown that absence of these cytokines of inflammatory nature does not affect MK development but rather acts in a synergic manner with TPO, stimulating MK maturation at the progenitor level [42,43,44]. However, it is important to acknowledge that adding some of these cytokines to the culture would create an inflammatory environment that directly affects megakaryopoiesis and its PLT produce [45]. None of the cells during this process are isolated entities but rather maintain a constant dialogue with their surroundings, taking up queues and reacting to them. Therefore, MK differentiation and PLTs produced under an inflammatory environment display a distinct phenotype compared to steady state conditions, as evidenced in recent studies of mouse models of sepsis [38]. In those cases, where there is high clearance and thus demand of PLTs, the conventional model of PLT production to replenish their number in the circulation is not sufficient. It is in this pathological situation that pro-inflammatory cytokines (such as IL-1α) induce a rapid MK rupture into PLTs, thus meeting that demand [46].

At present, there is still not a consensus on which factors or culture systems (soluble or matrix-based, static or dynamic) are needed in order to efficiently mimic physiological PLT production in vitro and to provide steady state competent PLTs at the functional level.

### 2.2. The Source Material and Its Developmental Stage

MK culture pursues the goal of understanding the process of megakaryopoiesis itself, not only to be able to faithfully mimic it but also to recapitulate a disease (phenocopy) in order to provide therapeutic solutions or phenotype rescuing [47]. Since the discovery of TPO, the scientific community has been able to successfully culture MKs and PLTs from a variety of HSC sources of both mice and human origin from fetal liver, umbilical cord blood (CB), and from both neonatal and adult peripheral blood (PB) and BM [48]. When using any of these sources, it is important to bear in mind the major developmental, phenotypical, and transcriptional differences in megakaryopoiesis between cells from fetal/neonatal and adult origin, which undoubtedly render PLTs with different phenotypes and functional characteristics that could have an impact in the final transfusion product [49]. Despite the experience acquired in the field from studies using mouse models, the development of in vitro culture methods for PLT production with cells of human origin has proven a difficult task, partially due to the difficulties to comprehensively characterize megakaryopoiesis in human [50,51].

During hematopoiesis, cells from fetal/neonatal sources yield larger numbers of small, low ploidy MKs, although mature in granule content [52]. This therefore leads to reduced levels of PLT production per MK. However, this seemingly negative liability balances out eventually due to the fact that HSCs and MK progenitors have a higher proliferative potential than their adult counterparts [53]. 

These and other phenotypical differences may reside in the composition of the niches at the various developmental stages and are due to the interplay between environmental factors [54], such as differential secretion of cytokines from stromal and sinusoidal endothelium cells from the fetal liver or osteoclasts and osteoblasts from the BM, distinctive responses to growth factors (included TPO), and cell-inherent mechanisms. As for the latter, in a study comparing the transcriptome at different stages of development, it showed that genes from fetal MKs were enriched in pathways such as angiogenesis, transforming growth factor β (TGFβ) signaling, and extracellular matrix (ECM), contrary to what would be expected, and emerged in adult MKs, that is, pathways related to PLT classical functions in hemostasis [52].

This shift towards non-conventional pathways was later translated into a production of PLTs that differed from the ones generated during steady state, adult megakaryopoiesis. Fetal/neonatal PLTs have shown noticeable different activation responses compared to adult PLTs that points to a hyporeactive phenotype in response to physiological agonists such as collagen, which is complemented by an enhanced response to mechanical stimuli and von Willebrand factor (vWF) receptor-mediated responses, together with the procoagulant properties of neonate blood [55]. Moreover, their transcriptional profiles display noticeable differences in protein metabolism, and little is known regarding the plethora of non-hemostatic functions assigned in recent years to PLTs and how this would influence PLT transfusion in infants/neonates vs. adults [56].

## 3. Human Cell Sources to Produce In Vitro Megakaryocytes and Platelets

The idea behind the production of PLTs in vitro demands a methodology of MK in vitro culture that takes advantage of a source material that guarantees proper differentiation and PLT production in sufficient numbers. However, there is still no agreement on this matter and, currently, different source materials are being employed, which we discuss below.

### 3.1. Hematopoietic Stem Cells and Precursors

The ex vivo production of PLTs focused first on the differentiation of primary human CD34^+^ HSCs, which have been successfully isolated from CB [57], PB [58] and, albeit rarely, from BM [32].

CB CD34^+^ cells, although limited due to their availability and authorized access, have been the most widely used source for MK culture in vitro. Subsequent developments and modifications in the growth conditions allowed for the culture of MKs from both PB and BM [59]. However, as it was noticed in the previous section, and in spite of presenting certain advantages compared to other HSC sources, differentiation of CB-derived MKs differ from their adult counterparts (e.g., PB or BM) in a way that may affect the functionality of the resulting PLTs [60,61].

PB CD34^+^ cells, although easy to obtain with non-invasive techniques and relatively minor processing, are still limited to ethical regulations and donations. While the relative frequency of CD34^+^ cells is lower than in CB samples [62], it can be enriched by G-CSF-induced mobilization of CD34^+^ progenitors from the BM to the PB [63]. In addition, other whole blood processed material may provide a more enriched CD34^+^ cell fraction, such as leuko-depletion filters and buffy coats from routine donations [64,65]. On this line, it has been shown that MKs can be differentiated from the human peripheral blood mononuclear cell (PMBCs) fraction without CD34^+^ cell enrichment (e.g., sorting). With this method, early hematopoietic precursors commit to the MK lineage in vitro given adequate growth conditions, and cells are unable to undergo this differentiation and eventually die in the culture [41]. This opens the possibility of avoiding cell sorting and thus reducing the costs of the process.

BM CD34^+^ cells, while they might represent the most appropriate source for recapitulating physiological MK differentiation from a given patient ex vivo, require invasive techniques for harvesting and seem unable to sustain the culture of MKs in large quantities and therefore may be used for other purposes than PLT production (e.g., disease-related basic or translational research) [33,60].

In any case, it is important to bear in mind that CD34^+^ sorted cells do not constitute a pure HSC population, however enriched. Furthermore, the purity of the starting material does not guarantee a pure, homogeneous, and synchronous culture, as mentioned before.

Lastly, although basic and translational research into megakaryopoiesis and PLT production in vitro is highly dependent on isolated HSCs, either from adult or fetal origin, and PBMCs, primary cells are not immortalized and thus have limited self-proliferation, making them unfit to meet the demands of the PLT number required for (a single) transfusion [66]. However, they can still be used, offering an inexpensive and accessible source, and more importantly, a source for rare phenotypes or compatible autologous stem cells.

### 3.2. Cell Reprogramming

The need to overcome the aforementioned limitations, namely self-renewal and PLT yield, drove some groups to develop cultures using the so-called human pluripotent stem cells (hPSCs), which comprise human embryonic stem cells (hESCs) [61] and the recently discovered human induced pluripotent stem cells (iPSCs) [67]. Both can be differentiated towards any cell type given the adequate growth conditions. In this case, it has been proven that they both are able to differentiate into MKs and produce PLT-like particles [68]. This directed differentiation via signals administered in the form of cytokine cocktails may lead to asynchronous cultures of suboptimal purity [69]. This means that some of these cells may persist after the differentiation has occurred, and given their embryonic nature, they have the potential to form teratomas [70]. However, although the collected PLTs can be irradiated to prevent remaining hPSCs from causing any damage, a large number of apoptotic cells and vesicles in the supernatant may elicit an immune response if transfused [71].

Other limitations associated with these cell sources are the ethical concerns raised by hESCs and the fact that both of them require sophisticated experimental setups that make them highly expensive to maintain, especially if we consider that the current PLT yields obtained from iPSCs, although high, are still not comparable to the amount of PLTs contained in a PC [72]. In addition, their culture still requires serum and feeder cells, which makes them unsuitable to be produced abiding cGMP guidelines [61,73]. Lastly, and more importantly, the impact of the developmental stage of these cell sources should not be overlooked, as commented upon before, since the use of both hESCs and iPSCs involves an embryonic hematopoietic status when reprogramming cells towards MKs.

Despite all of the above, iPSCs are widely regarded as the best solution for the scientific community to solve the problem of producing ex vivo, cGMP-ready PLT transfusions [74]. They present with advantages that outweigh their drawbacks, such as being an inexhaustible, self-renewable source of MKs and PLTs [72]. They can also be genetically modified to, for example, match any major histocompatibility group or, contrariwise, to be devoid of HLA antigens to give rise to universal PLTs, minimizing the risk of refractoriness and alloimmunization [75]. In addition, studies have shown that it is possible to drive these cultures using feeder- and serum-free mediums [76,77]. As for concerns regarding culture purity, ongoing efforts such as forward programming seem to be closing in what mechanisms might drive fate decisions in MK differentiation [78].

Lastly, although currently a seemingly viable option, iPSCs still have a long way to go in terms of proving their capacity to produce fully functional PLTs with the right morphology, surface receptors, and adequate half-lives in sufficient numbers and at accessible costs.

### 3.3. Non-Hematopoietic Sources

Fibroblasts, as well as endothelial and adipose tissue-derived stromal cells (ASCs) have been used as source material to generate MKs in vitro. Endothelial cells can be driven towards MK differentiation through an HSC transition stage upon adequate stimulation [79]. Likewise, fibroblasts, which also express MPL, can be coerced into becoming MKs when transfused with certain factors [80]; however, ASCs seem to perform better overall. In a first approach, a study conducted by Ono-Uruga et al. found a simple and cost-effective way to differentiate ASCs to MKs by means of endogenous TPO, and without genetic manipulation or feeder cells [81]. Resulting PLTs were found to be functional, with normal PLT surface marker expression and in numbers comparable to those obtained with iPSCs. The major drawbacks present in this study were that only a subpopulation of ASCs was able to differentiate into a MK lineage, and that they were non self-renewable. Nonetheless, a recent study by the same group aimed to overcome said limitations by developing an ASC line (ASCL) and thus turning the original source into a donor-independent one [82]. However, although similar in hemostatic function to in vivo PLTs, ASCL-derived ones were found to be slightly hyperreactive (as opposed to the PLTs produced from ASCs), and the fact that it is not an immortalized cell line would make it necessary to produce new ones regularly, which in turn would result in inter-batch variability.

The developmental stage of the source material is not the only parameter that might influence the characteristics of the final PLT produce. The choice and the possibilities to develop PLT production culture methods from either primary or reprogrammed cells must be taken into account, for which we should know and describe their differentiation characteristics and functional properties of the produced PLTs.

## 4. The Culture Engineering

Another aspect of concern regarding in vitro PLT production that should not be overlooked is the physical characteristics of the culture. Currently, PLT yields obtained in vitro are far from those encountered in physiological conditions, and it is mainly due to an inefficient release from in vitro cultured MKs, which adds to the lack of synchronicity and controllable MK terminal differentiation. Ongoing efforts are focusing on overcoming this bottleneck by developing bioengineering techniques that would reproduce the complex interactions that occur within the BM [83,84]. Whether the culture is performed in a static or a dynamic system or by implementing a three-dimensional (3D) matrix or scaffold is a matter of discussion, and developments are being done in this regard.

Adult hematopoiesis occurs in the BM, a complex 3D environment that exerts constraints on the development of the resident cells, and thus the architectures of the stromal niche, the endothelium, and the circulation are determinant in healthy PLT production [85,86]. Although the dynamics of such an environment are still poorly understood, efforts must be made to distance ourselves from the classical suspension culture and aim towards mimicking, as closely as possible, the physiological (soluble factors, cell types) and the physical (stiffness, rheology) aspects that characterize the BM niche [87]. Recreating such an environment has proven to improve in vitro MK differentiation and proplatelet formation [88].

Lastly, other added difficulties stemming from the in vitro culturing of MKs are caused by the heterogeneity, despite the enrichment of MKs, and the asynchronous nature of MK cell differentiation, plus other events such as cell death, vesiculation, etc. This makes it difficult to harvest the produced PLTs all at once while remaining pure, free of debris and vesicles, and maintaining their functional integrity. Current methods developed to tackle this issue and to separate PLTs from MKs, progenitors, or debris include serial centrifugations at different time points [63]. This technique, however, might result in yield reduction, PLT activation, and it is time-consuming and difficult to standardize to meet cGMP requirements. Recently, a procedure involving a spinning-membrane filtration device was developed that ensures the return of a pure population of PLTs with high yield and maintains PLT functional integrity [89]. Furthermore, contaminating MKs were able to resume proplatelet formation and PLT production when returned to culture conditions.

### 4.1. Bioreactors

Ex vivo PLT production has come a long way since MKs were first cultured, and bioreactors represent state-of-the-art technology aimed towards the goal of manufacturing PLTs in sufficient numbers to replace donor-derived PLTs [90]. They have the advantage of being easily adaptable and of functioning independently of the cell source used. Most of them can also be scaled-up, to a greater or lesser extent, to accommodate large numbers of MKs and thus be commercially and clinically suitable.

In the last years, the main effort to recapitulate the BM environment has focused on developing microfluidic bioreactors to faithfully mimic key physiological characteristics, such as the ECM composition, BM stiffness, soluble factors, and blood vessel architecture, which includes tissue-specific microvascular endothelium, endothelial cell contacts, and circulatory shear stress [91]. They constitute the most suitable method for large-scale production due to its scalability, handling, control of cell density, uniform nutrient distribution, and of culture conditions (e.g., pH, temperature, and O_2_ and CO_2_ concentrations) [92,93].

When MKs are exposed to a microenvironment that reproduces the essential characteristics of their native niche, it is expected that MK maturation, proplatelet formation, and shedding of PLTs will occur appropriately. When switching from 2D to 3D cultures, there is an increment of the surface area, which in turn allows for more interactions between MKs and their proplatelets and the surrounding co-cultured cells or matrix scaffolds. To achieve this kind of environment, several scaffolds have been used, made of hydrogel [94], polyester, or polydimethylsiloxane (PDMS) [95], coated with ECM proteins (e.g., fibronectin, collagen, vWF), and perfused with media containing specific growth factors and cytokines [91].

A major concern regarding these scaffolds is the biocompatibility and the scalability of the materials. Di Buduo et al. first aimed at tackling this issue by using a silk fibroin sponge [96], which was further developed by designing a microtube-like structure made of PDMS, coated with such a sponge [97]. This structure was also covered on the inside by endothelial cells, vascular endothelial growth factor (VEGF), and other ECM-mimicking components. At the same time, they considered other factors that affected the rheology of this novel material, such as stiffness, and introduced a flow composed of red blood cells to modulate viscosity and shear stress [97].

On this line, several studies have shown the importance of an optimal control of the physical attributes of the bioreactor. While hypoxia and hypothermia may help in the expansion and the differentiation of MKs [92,98], respectively, it has been shown that control of the flow rate (i.e., shear stress) favors rapid proplatelet formation and PLT production [99]. On this line, Thon et al. manufactured a chip-based microfluidic bioreactor that, apart from demonstrating the optimal rate of shear stress by using parallel flows, also incorporated certain aspects of the BM environment, such as stiffness, pore size, ECM composition, and endothelial cell contacts [100]. Other approaches that build on this attribute in order to control PLT release are a microfluidic device that retains MKs in vWF-coated micropillars [101], developed by Blin et al., and a two-chamber nanofiber membrane-based bioreactor with two perpendicular flows [102], developed by Avanzi et al. However, there is still no consensus on whether it is best to use dual or single flows to increase production [103] and whether a confluent system where different flow angles correlated with different PLT yields may be more effective [95].

Lastly, Ito et al., prompted by the long-lasting issue of the insufficient PLT numbers generated by bioreactors, have recently identified turbulence as a key physical factor in the management of large-scale PLT production [104]. By conducting BM in vivo imaging and particle image velocimetry, they observed that turbulent forces were only present around proplatelet-carrying MKs and thus were able to simulate this condition in a bioreactor and, by means of scaling-up, to produce near-clinical numbers of PLTs.

However, a deeper characterization of the PLT produce is necessary in order to assure its interchangeability with their donor-derived counterparts [105]. Once PLTs are obtained, they must be concentrated, purified, washed, and suspended in a preserving solution compatible with transfusion, which represents a challenge due to the asynchronous and the heterogeneous nature of these cultures. Furthermore, this enrichment is especially important to avoid debris, microvesicles, and other particles that carry some resemblance to PLTs, but are not able to perform their usual tasks, although they might be capable of adhering to certain substrates, thus potentially impoverishing the quality of the product.

Although some groups have characterized, to a greater or lesser extent, the PLTs generated under their specific culture conditions, a consensus on what to do exactly is still lacking [96,104,106]. We propose, first, a PLT standardization protocol encompassed within the production method, consisting of a series of functional and characterization tests, with the aim of determining that the final products are, in fact, viable PLTs that exert their expected function (e.g., immunophenotyping, viability tests, morphological and molecular characterization, adhesion, degranulation, and aggregation assays) (see Table 1). This, in turn, will provide insights into the manufacturing process, allowing its targeted improvement. Secondly, once a standard procedure is consolidated, a selection of assays could be settled for the quality control of every subsequent manufactured batch to assure the quality and the function of the in vitro derived PLTs (see Table 1 and Figure 1).

As an additional consideration, the production method should be as brief as possible, given that the in vitro produced PLTs would face the same storage limitations as their donor-derived counterparts, and should be readily available in cases of emergency. The quality of the product through storage should also be determined. Lastly, one significant issue that would undoubtedly hinder the efforts towards its widespread use is the extremely high production cost per unit. Laboratory-generated PLTs ultimately must be priced competitively relative to donor-derived PLTs in order to be effectively implemented in a clinical setting.

### 4.2. Using the Lungs as In Vivo Bioreactors

Another organ, besides the BM, where MKs have been shown to reside and shed PLTs, is the lung [108]. Thus, an alternative and already tested approach that would abolish the need to produce in vitro PLTs is turning the lungs into in vivo bioreactors by intravenously infusing ex vivo cultured MKs and using the pulmonary bed as the location of PLT production. In a first approach, murine MKs of BM or fetal liver origin were infused into mice and shown to be able to release functional PLTs [109]. In a second approach, a similar experiment was performed, but in this case, human CD34^+^ cells from the BM were xenoinfused into immunodeficient mice, which became entrapped in the pulmonary microvasculature [110]. A small percentage was also found in the spleen, while there were no signs of entrapment in either the BM, the liver, the heart, or the brain. PLTs released under these circumstances were comparable to donor-derived PLTs in terms of size, granule distribution, half-life, and surface marker expression but were released in a delayed manner and were rapidly cleared from the circulation. As for their functionality, although similar, they showed reduced convulxin (CVX) responsiveness and incorporation to thrombi [110].

However, this approach, albeit promising, poses its own risks that may outweigh its benefits, since the number of mature MKs that would have to be injected to achieve the PLT numbers provided by an apheresis unit could potentially cause capillary obstruction, and their extruded nucleus may elicit inflammation and autoimmune responses [110]. Another issue would be the time it takes for these transfused MKs to produce PLTs (approximately 24 h) versus direct infusion of PLTs. In addition, if genetically modified iPSCs were to be used as source, the risk of tumorigenesis would have to be addressed, although it has been shown that radiation of MKs might not affect either proplatelet formation and shedding or PLT functionality [111,112]. Lastly, these studies were performed in murine models, and thus any conclusion as to whether this could be a viable option in humans would have to wait, although their safety and their tolerability have already been tested in phase I clinical trials with positive results [113].

### 4.3. Artificial Platelets and Applications beyond Transfusion Medicine

The development of artificial PLTs seems a promising alternative for the acute bleeding patients requiring immediate transfusions, as these hybrid synthetic PLTs are compatible with all blood groups, comply with transfusion safety requirements, are biodegradable, easy to manufacture, allow for longer storage lives, and their production costs are more reduced than those of in vitro PLT production (considering a single application unit) [114]. The ultimate goal of these artificial PLTs is to recreate their functions of adhesion and aggregation as well as to mimic their morphology and physical properties, therefore overcoming the aforementioned limitations in the obtention of physiological-like PLTs.

There are currently several approaches to the manufacture of these pseudo-PLTs, such as the thrombosomes and the infusible PLT membrane, which make use only of the membrane and its associated receptors [115,116]. Other methods consist of adorning the surface of albumin microparticles, red blood cells, or synthetic polymers with fibrinogen or fibrinogen-mimetic peptides [117,118,119]. More recent developments include the use of the latter in combination with vWF- and collagen-binding peptides to coat the surface of liposomal nanoconstructs [120].

Nonetheless, the use of in vitro-generated PLTs might extend beyond their application in transfusion medicine. PLTs and artificial PLTs can be converted into drug carriers, given their ability to target and migrate to sites of injuries or disease [121,122,123]. In addition, their newly-identified functions in immune modulation, inflammation, cancer, and tissue regeneration [124,125,126] open the door to the in vitro production of fine-tuned PLTs, both at the receptor and the cargo levels, that can potentially modulate these pathological states [127,128,129].

## 5. Conclusions

In recent years, there have been enormous advances towards the manufacturing of PLTs ex vivo in order to meet the rising demand of PLT transfusions and to be less dependent on donations. Such advances have come from both the field of the in vitro culture of MKs, in terms of cell sources and cytokine cocktails, and the developments regarding growth conditions that mimic the BM niche (Figure 1). Bioreactors coupled with immortalized cell lines, namely iPSCs capable of undergoing megakaryopoiesis, might be the closest the scientific community has gotten to replicating the process ex vivo, with PLT yields similar to those found in PCs from donations (Figure 2).

Despite these advances, there are still hurdles that must be overcome. Costs and production times are too high (up to 26 days), and added to the fact that their shelf life is the same as their donated counterparts, this makes them unsuitable in cases of emergency, where large numbers of PLTs are needed in a very short period of time. Furthermore, a more thorough characterization of both MKs and PLTs produced in vitro is required [130]. Many studies have focused on the PLT yield while obviating either their morphology, their protein surface expression, or their functionality both in vitro and in vivo. On this line, these ex vivo PLTs have yet to be fully tested in humans. Clinical trials have only focused thus far on their safety and tolerability, meaning further trials to look into efficacy, function, and other specific variables must be performed. It is also important to distinguish between those patients where a PLT transfusion is essential and those where TPO administration or alternative methods, such as artificial PLTs, might suffice.

Considering as well the natural PLT distinct functional profiles due to the developmental stage (infant vs. adult) or the health status of an individual (healthy or with subjacent inflammation), it opens the question as to whether in vitro produced PLTs from whichever source, method, and conditions should be suitable for a given specific patient and clinical circumstances. Furthermore, we lack knowledge on whether in vitro produced PLTs may be prone to acquiring PLT storage lesions in the same way as donor PLTs, whether they can be stored in the same conditions and for the same time, and most importantly, the question arises as to how we set production protocols to not only abide cGMP regulations but also, regarding transfusion safety, whether these in vitro produced PLTs require PRTs and how that would affect their integrity.

Lastly, there are currently a plethora of protocols regarding MK differentiation and PLT production. They all differ in the starting cell source, growth conditions, and physical forces applied within the 3D culture, and while this variety of methods has greatly aided in the refinement of every step of megakaryopoiesis, it also makes it difficult to reach a consensus on what protocol to use. Paving the way to standardization will position ourselves closer to production of cGMP-grade PLTs and their final implementation in the clinical practice.

## Figures and Tables

**Figure 1 medicina-56-00671-f001:**
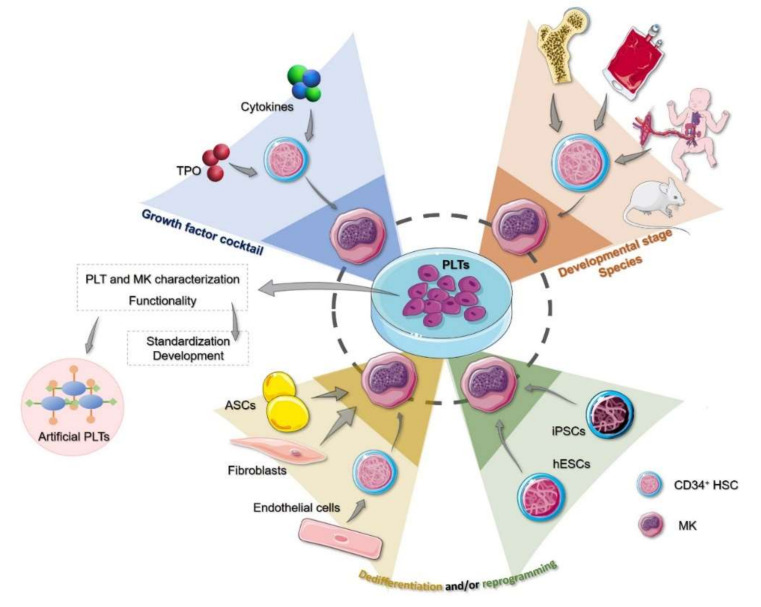
Variables conditioning the in vitro culture of MKs for PLT production: growth factors and cellular sources. Blue triangle: the appropriate growth factor cocktail aims at the differentiation of precursors into mature MKs, however, there is no consensus on the cocktail composition. Orange triangle: the source of hematopoietic precursors used to date are from adult or fetal origin, human or mouse, however, while acknowledged, we sit at the tip of the iceberg regarding the comprehensive knowledge of the differences on megakaryopoiesis and PLTs derived from these different sources. Green and yellow triangles: reprogramming and dedifferentiation approaches. iPSCs or hESCs may provide advantages to primary cultures due to their immortality, while they present ethical concerns (green triangle). Other approaches include dedifferentiation and/or reprogramming of non-hematopoietic cells, such as endothelial cells, ASCs, and fibroblasts (yellow triangle). No comprehensive characterization of megakaryopoiesis and PLTs has been done on these models. Red circle: artificial PLTs constitute a viable option only in certain conditions (e.g., trauma). In summary, before any implementation in a clinical setting, a full characterization of both MKs and PLTs must be performed to ensure the correct functionality and safety of this product. This characterization will, at the same time, contribute to the standardization and the improvement of culture methods. TPO, thrombopoietin; MK, megakaryocyte; PLT, platelet; HSC, hematopoietic stem cell; hESC, human embryonic stem cell; iPSC, induced pluripotent stem cell; ASC, adipose tissue-derived stromal cells.

**Figure 2 medicina-56-00671-f002:**
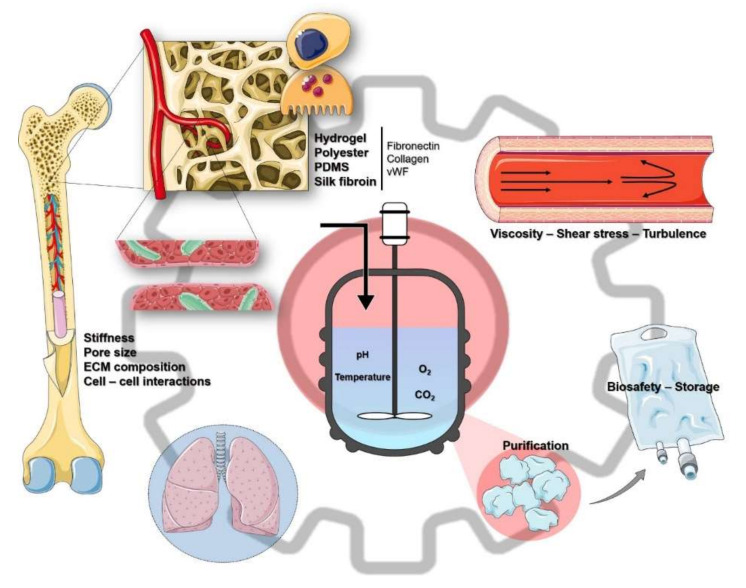
The culture engineering. Bioreactors aim to mimic the complex 3D structure and interactions that occur within the human BM. For this purpose, it is important to control the physical characteristics of the BM (e.g., stiffness, cell–cell interactions) by using biocompatible materials (e.g., PDMS, silk fibroin, coated with collagen) and recapitulating its architecture, including the characteristics of the blood vessels (i.e., rheology). The resulting PLTs have to be isolated from the rest of the culture and stored until their use under the proper conditions, thus their safety is guaranteed and their functionality untouched. Lastly, it has been shown that the lungs may constitute another source of PLTs when MKs become trapped in the microvasculature and could be used as in vivo bioreactors, transfusing MKs that produce PLTs after homing to the lungs. ECM, extracellular matrix; PDMS, polyldimethylsiloxane; vWF, von Willebrand factor.

**Table 1 medicina-56-00671-t001:** Proposed characterization and functional PLT tests that would comprise the standardization phase and/or the quality control per se. WB, western blot; ELISA, enzyme-linked immunosorbent assay; qRT-PCR, real-time quantitative reverse transcription polymerase chain reaction.

	Standardization Phase	Quality Control
**PLT characterization**	Immunophenotyping–comprehensive receptor expression (flow cytometry)	Immunophenotyping selected receptors
	Morphology (size, granularity; flow cytometry, e-microscopy)	Morphology (flow cytometry)
	Viability/Apoptosis tests (mitochondrial function, caspase activity) [107]	If relevant from standardization, mitochondrial function
	Proteomics	If relevant from standardization, selected proteins (WB, ELISA, Luminex)
	Transcriptomics	If relevant from standardization, selected transcripts (qRT-PCR)
**PLT function**	Adhesion (comprehensive substrate panel)	The ideal would be to minimize functional tests, when the characterization reaches a certain quality level as derived from the standardization phase
	Degranulation (comprehensive agonist panel)
	Aggregation (comprehensive agonist panel)	Aggregation (selected agonists)
	Thrombi formation (perfusion assays in vitro)	
	In vivo transfusion assays

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
