# Peer review of "On the Quest for In Vitro Platelet Production by Re-Tailoring the Concepts of Megakaryocyte Differentiation"

_medicina, 2020, doi:10.3390/medicina56120671_

Round 1

Reviewer 1 Report

The manuscript is an excellent review of in vitro platelet production based on megakaryocyte differentiation. It regards very important problem of platelet transfusion, a worldwide increase in the demand of platelet concentrates from human donors and an evolution around the in vitro differentiation of megakaryocytes into platelets. The Authors summarize advantages and disadvantages of an in vitro or an ex vivo platelet production paying attention on its potential usefulness in clinical application and current Good Manufacturing Practice. The manuscript is prepared well, it includes two schemas illustrated the in vitro culture of megakaryocytes and the culture engineering.

In my opinion, a great addition to paper could be a paragraph describing morphology, functionality and hemostatic properties of cultured platelets which are to be considered as a clinical alternative of platelet concentrates to transfusion. The publication of Do Sacramento et al. is one from the articles reported this issue (Do Sacramento V, Mallo L, Freund M, Eckly A, Hechler B, Mangin P, Lanza F, Gachet C, Strassel C. Functional properties of human platelets derived in vitro from CD34+ cells. Sci Rep. 2020 Jan 22;10(1):914. doi: 10.1038/s41598-020-57754-9).

I also suggest the revision of some keywords, such as development, in vitro culture.

Author Response

We thank the reviewer for their suggestions. We agree that an extra paragraph regarding the full characterization of the cultured platelets is necessary, and so we have added that new information to the text, page 15 (lines 357 to 385), supported by a table (Table 1, page 17). Additions are marked in red font and highlighted in yellow and deletions are shown as strikethrough text. Briefly, although most of the studies focusing on the in vitro production of platelets for transfusion purposes perform a characterization of said platelets, they rarely overlap, and result in partial analysis of the functional capacity and cytological/molecular characteristics of platelets. We now propose, first, a standardization phase aimed at the comprehensive characterization of the produced platelets. This will allow to both improve the production methodology, and to select the relevant assays for the established quality control when a method is consolidated. Table 1 summarizes it.

We have included the reference suggested by the reviewer (now ref. 106).

We have removed the keywords pointed out by the reviewer, and substituted them with “in vitro platelet production” and “cell source”, which we would modify in case the reviewer finds them still not appropriate.

Reviewer 2 Report

This review article overviewed the history of studies regarding megakaryopoiesis and subsequent thrombopoiesis in vivo and ex vivo (in vitro).

Although the authors do not discuss the heterogeneity of hematopoiesis and megakaryopoiesis, they stated how somatic blood system can be applied to ex vivo manufacturing platelets.

Overall story is well written.

Author Response

We thank the reviewer for their comments. Regarding the heterogeneity of hematopoiesis and megakaryopoiesis, that is precisely what we discuss when referring to the source material (species, developmental stage of the hematopoietic progenitor source, iPSCs, etc.; section 2.2, section 3), growth factor cocktail required for megakaryopoiesis (no consensus, section 2.1) and how heterogeneous and asynchronous the cultures are (page 13, line 294). We agree that this is discussed briefly, as the scope of this review is to highlight these points of concern, so they are taken into account when a method is developed.